# TACO: Think-Answer Consistency for Optimized Long-Chain Reasoning and Efficient Data Learning via Reinforcement Learning in LVLMs

## Abstract

The paradigm for training Large Vision-Language Models (LVLMs) is evolving toward autonomous problem-solving, revealing critical instabilities in complex visual reasoning. We identify three failure modes: exploration collapse, inefficient learning, and—most notably—ineffective reasoning, marked by logical inconsistencies between reasoning traces and outputs. To mitigate these, we introduce TACO, a reinforcement learning framework that enforces multi-level consistency. TACO comprises three integrated components: a Think-Answer Consistency (TAC) reward ensuring joint alignment of reasoning, answer, and ground truth for semantic integrity; a Memory-Guided KL Stabilization (MKS) mechanism that dynamically defers high-risk updates to prevent optimization collapse; and an Adaptive Difficulty Sampling (ADS) module that optimizes data curation for efficient learning. Extensive experiments validate TACO's superiority, achieving top performance on 15 benchmarks spanning Referring Expression Comprehension (REC), Visual Question Answering (VQA), and long-horizon Video VQA. TACO exhibits enhanced generalization, sustained efficiency, and stability in long-chain reasoning, outperforming conventional RL approaches.

## 1 Introduction

The training paradigm for Large Vision-Language Models (LVLMs) (Xu et al., 2016; Agrawal et al., 2016) is evolving from imitative learning, such as Supervised Fine-Tuning (SFT) and Reinforcement Learning from Human Feedback (RLHF) Achiam et al. (2023); Bai et al. (2025); Lu et al. (2024); Chen et al. (2024b), toward an autonomous problem-solving approach. This new paradigm, exemplified by Reinforcement Learning with Verifiable Rewards (RLVR) Xiao et al. (2025), aims to enhance complex reasoning by shifting the learning objective from imitation to solution-oriented achievement. However, directly applying this paradigm to the complex scenario of visual long-chain reasoning exposes a series of severe challenges. These include the classic problems of "exploration collapse", "inefficient learning", as well as a more deceptive failure mode that we identify as "ineffective reasoning", whose core pathology lies in the logical inconsistency between the model's reasoning process and its final output.

While prior works treat these as isolated bottlenecks, we argue that they arise from a systemic breakdown in consistency across semantic, optimization, and learning levels. To address this, we propose the TACO framework, which enforces multi-level consistency to construct an efficient and stable learning process. TACO extends the foundational RL concept that constraints bring stability Schulman et al. (2015; 2017) from a singular optimization focus to an integrated framework covering all three levels:

- **Semantic Consistency.** A key challenge in complex reasoning is *Ineffective Reasoning*, where a model produces a correct final answer from a flawed or irrelevant reasoning process. To combat this issue and enforce semantic integrity, TACO introduces the Think-Answer Consistency (TAC) reward mechanism to provide hierarchical supervision by mandating a joint alignment among the

Figure 1: Overview of the TACO framework. It systematically resolves the disorder in LVLM training by synergistically managing three consistency modules: **Learning Consistency (ADS)** for efficient data sampling, **Optimization Consistency (MKS)** for training stability, and **Semantic Consistency (TAC)** for coherent reasoning, ultimately achieving a significant performance advantage over the baseline (right).

reasoning chain ($Think$), the final answer ($Answer$), and the ground truth ($GT$), which compels the model to learn a valid and coherent thought process.

- **Optimization Consistency.** Generating long, precise reasoning chains often triggers exploration collapse. This occurs when the policy sharpens, causing a KL divergence spike and a catastrophic gradient imbalance where the stability penalty dominates the reward signal. TACO's Memory-Guided KL Stabilization (MKS) mechanism acts as a dynamic regularizer, using an adaptive threshold to identify and defer high-risk samples via an experience buffer, thus preventing destructive updates and ensuring training stability.

- **Learning Consistency.** To proactively prevent such instability and address inefficient learning, the Adaptive Difficulty Sampling (ADS) module curates the training data stream. Inspired by the Zone of Proximal Development (ZPD), ADS dynamically prioritizes samples that are challenging yet achievable. This strategy maximizes the efficiency of each gradient update, accelerating convergence by ensuring the model is optimally engaged.

In summary, we propose TACO, a reinforcement learning framework for visual reasoning in Large Vision-Language Models (LVLMs), which enforces multi-level consistency to reframe training as a stable, efficient, goal-oriented process. Our main contributions are threefold: **1)** introducing TACO to unify key challenges as three consistency failures, including semantic, optimization, and learning levels; **2)** designing three synergistic mechanisms (TAC, MKS, and ADS), where TAC enforces semantic consistency through joint geometric and semantic alignment; and **3)** achieving state-of-the-art performance on 15 in-domain and out-of-domain REC and VQA benchmarks using different foundation models, demonstrating superior generalization, sustained learning efficiency, and stable training in long-chain reasoning where traditional RL methods falter.

## 2 RELATED WORK

**Large Vision-Language Models (LVLMs).** LVLMs bridge vision and language. Core advances include large-scale contrastive pre-training for joint embeddings (e.g., CLIP Radford et al. (2021)) and LLM-style instruction tuning for enhanced visual dialogue/reasoning (e.g., LLaVA Liu et al. (2023a)). Dealing with varied image sizes is key. Dynamic methods (AnyRes Chen et al. (2024b); QwenVL techniques Bai et al. (2023)) aid input flexibility. However, complex reasoning and generalization remain tough.

**Reinforcement Learning (RL) in LVLMs.** RL offers a compelling way to enhance reasoning, building on language successes like RL's efficacy on logical tasks OpenAI (2024) and GRPO enabling direct reasoning optimization (potentially bypassing SFT, DeepSeek-R1 Guo et al. (2025)). For multimodal RL, however, addressing cross-modal consistency and stability is key. Efforts in this area include developing specialized reasoning datasets with formalized visual inputs (R1-OneVision Yang et al. (2025)), successfully porting RL algorithms like GRPO to VLM training (R1-V, Visual-RFT, VLM-R1 Chen et al. (2025); Liu et al. (2025); Shen et al. (2025)), and introducing mechanisms

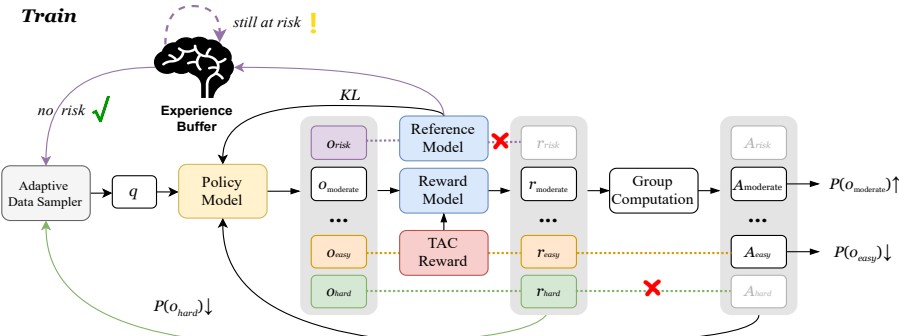

Figure 2: Overview of the TACO training loop. This figure illustrates the synergistic workflow of the three core components: ADS, MKS, and TAC within a single training step. The ADS (left) manages input sampling, the Policy Model (center) performs exploration, and the MKS (top) identifies and isolates high-risk outputs ($o_{\text{risk}}$) via an Experience Buffer and KL divergence monitoring, ensuring that only stable and effective signals are used for the final policy update.

like verifiable rewards Liu et al. (2025). Intriguingly, applying RL directly to base VLMs has been shown to induce significant performance jumps or "visual epiphanies" (VisualThinker-R1-Zero Zhou et al. (2025)), with related work observing correlations between response characteristics like length and reasoning improvements under RL optimization (MMEureka Meng et al. (2025)). While VLM-R1 Shen et al. (2025) shows the potential of RL, it also reveals challenges like reasoning inconsistencies and model instability, which our work aims to address.

## 3 METHOD

TACO framework manages the LVLM reinforcement learning process through a synergistic closed-loop system, as illustrated in Fig. 2. The core logic of it begins with Semantic Consistency: the Think-Answer Consistency (TAC) reward establishes a semantically correct objective for the model's complex reasoning. To ensure the stability of the high-risk exploration toward this objective, the Memory-Guided KL Stabilization (MKS) mechanism isolates risky explorations via an experience buffer, enforcing Optimization Consistency. Finally, to enhance the efficiency of the entire learning process, the Adaptive Difficulty Sampler (ADS) optimizes data input to achieve Learning Consistency, thus providing the most efficient data fuel for the system's stable operation.

### 3.1 PRELIMINARY

**Group Relative Policy Optimization**  Group Relative Policy Optimization (GRPO) enhances PPO Schulman et al. (2017) by eliminating the critic component. For input $x$, GRPO samples $N$ responses $\{o_1, o_2, \ldots, o_N\}$ from policy $\pi_\theta$, computing rewards $r_i = R(x, o_i)$. Relative performance is evaluated using advantage values $A_i$, which standardizes rewards without requiring a separate value function. The policy $\pi_\theta$ is updated by optimizing the GRPO objective:

$$J_{\text{GRPO}}(\theta) = \mathbb{E}_{\{o_i\}_{i=1}^N \sim \pi_{\theta_{\text{old}}}(x)} \left[ \frac{1}{N} \sum_{i=1}^{N} \{\min(s_1 \cdot A_i, s_2 \cdot A_i)\} - \beta D_{\text{KL}}[\pi_\theta \| \pi_{\text{ref}}] \right], \quad (1)$$

where $s_1 = \frac{\pi_\theta(o_i|x)}{\pi_{\theta_{\text{old}}}(o_i|x)}$ and $s_2 = \text{clip}\left(\frac{\pi_\theta(o_i|x)}{\pi_{\theta_{\text{old}}}(o_i|x)}, 1 - \epsilon, 1 + \epsilon\right)$. The term $D_{\text{KL}}[\pi_\theta \| \pi_{\text{ref}}]$ represents the Kullback-Leibler divergence between the current and reference policies, weighted by $\beta$.

**Referring Expression Comprehension and Visual Question Answering.**  Referring Expression Comprehension (REC) is a multimodal task enabling machines to localize target objects or regions within a visual scene based on a natural language expression Qiao et al. (2020). Unlike traditional object detection, REC requires complex instructions and enhanced visual perception, making it valuable for applications like human-centric scenarios, autonomous driving, and medical image analysis He et al. (2023). Visual Question Answering (VQA) tasks, in contrast, involve generating

accurate natural language answers based on an image and a question Antol et al. (2015); Srivastava et al. (2021). Successful completion of VQA demands capabilities in object recognition, attribute understanding, and relational analysis. We extend experiments on these tasks to validate the robust reasoning and generalization capabilities of TACO.

### 3.2 TAC: A Hierarchical Supervision System for Semantic Consistency

While existing works employ Long Chains of Thought (Long CoT) to boost the reasoning ability, they risk the *ineffective reasoning* problem: generating a correct answer from a flawed or irrelevant reasoning process. Standard rewards on the final output fail to penalize such logical fallacies. Unlike standard outcome-based rewards that allow reward hacking Cobbe et al. (2021), we introduce the **Think-Answer Consistency (TAC)** reward, enforcing joint alignment among the reasoning chain ($Think$), final answer ($Answer$), and ground truth ($GT$). The general form of the TAC reward is expressed as:

$$R_{TAC} = f(Think, Answer, GT), \tag{2}$$

where $f$ is a task-specific metric function evaluating the alignment among $Think$, $Answer$, and $GT$. While existing rewards only focus on the final output, leading to ineffective reasoning, TAC addresses this issue through hierarchical supervision.

**REC Task Instantiation**  In the Referring Expression Comprehension (REC) task, to explicitly decouple reasoning and the final answer, we use a specific prompt format to guide the model to output the reasoning process within `<think>` tags and the final localization within `<answer>` tags. We instantiate the joint alignment principle as a geometric constraint on three bounding boxes: the reasoning process ($Think_{BBox}$), the final answer ($Answer_{BBox}$), and the ground truth ($GT_{BBox}$). The TAC reward is directly defined by their Intersection over Union (IoU):

$$\begin{aligned} R_{acc}^{REC} = R_{TAC}^{REC} &= IoU(Think_{BBox}, Answer_{BBox}, GT_{BBox}) \\ &= \frac{Area(BBox_1 \cap BBox_2 \cap BBox_3)}{Area(BBox_1 \cup BBox_2 \cup BBox_3)}, \end{aligned} \tag{3}$$

This single metric simultaneously evaluates accuracy and consistency, as any deviation among reasoning, answer, or ground truth reduces the reward, effectively incentivizing the model to learn a reliable localization reasoning process. Additionally, we include a format reward $R_{format}$, defined as (`<think>...</think><answer>...</answer>`), to encourage adherence to the specified output structure. Thus, the total reward for the REC task is $R^{REC} = R_{TAC}^{REC} + R_{format}$.

**VQA Task Instantiation**  For Visual Question Answering (VQA) tasks, where the answer space is discrete text, we employ an external supervisor model $S$ (e.g., Qwen-32B in our experiment or a domain-specific evaluator) to programmatically assess semantic consistency. Given a question $Q$ and ground truth $GT$, the supervisor $S$ evaluates whether the model-generated reasoning process $T$ logically supports an answer consistent with $GT$. The output score serves as the TAC reward for VQA:

$$R_{TAC}^{VQA} = S(Q, T, GT), \tag{4}$$

The total reward for VQA, $R^{VQA} = R_{TAC}^{VQA} + R_{acc}^{VQA} + R_{format}$, comprises three components: the core TAC reward $R_{TAC}^{VQA}$, which leverages the supervisor's judgment to suppress speculative answers; the conventional accuracy reward $R_{acc}^{VQA}$, which provides feedback on correctness (1 or 0 for closed-ended tasks, or edit distance for open-ended tasks); and the format reward $R_{format}$, ensuring adherence to the output structure.

### 3.3 MKS: A Dynamic Regularization Mechanism for Proactive Gradient Imbalance Management

Generating effective Long CoT is critical for complex visual reasoning, yet it exposes a fundamental instability in the training process. The experimental results in Fig. 3 clearly reveal this core challenge: while our TAC mechanism (orange line) successfully incentivizes the exploration of more effective Long CoT compared to the baseline (purple line), it leads to a fatal collapse. After a brief period of growth, an explosive spike in KL divergence (Fig. 3c) occurs in precise synchrony with the collapse

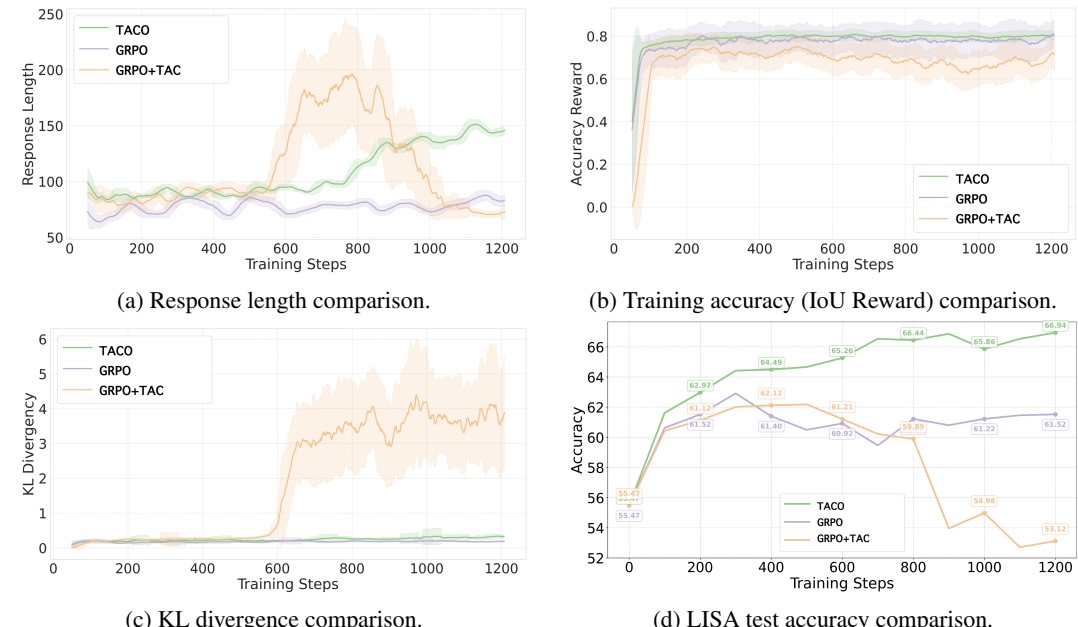

(a) Response length comparison.

(b) Training accuracy (IoU Reward) comparison.

(c) KL divergence comparison.

(d) LISA test accuracy comparison.

Figure 3: Effectiveness of the Think-Answer Consistency (TAC) reward, comparing TACO, GRPO, and GRPO + TAC. The subplots illustrate TAC's influence on: (a) response length evolution; (b) training accuracy (IoU reward) and the critical reasoning-answer alignment; (c) policy stability, tracked via KL divergence; and (d) Performance on the LISA test set.

of all performance metrics (Figs. 3a, 3b, 3d). Our analysis pinpoints the cause of this collapse to **Optimization Inconsistency**, an extreme gradient imbalance between reward maximization ($g_{\text{reward}}$) and constraint ($g_{\text{KL}}$) in the GRPO objective. The push to generate coherent Long CoT triggers policy entropy collapse—a rapid sharpening of the policy distribution. This, in turn, causes the KL divergence value to soar, making its corresponding gradient, $\|g_{\text{KL}}\|$, dominate the reward gradient.

To address the issue, we design the **Memory-Guided KL Stabilization (MKS)** mechanism, which identifies high-risk samples via a dynamic threshold $\delta$ and utilizes an experience buffer for their isolation and periodic re-evaluation. MKS hinges on an adaptive threshold, $\delta$, that scales the KL divergence tolerance based on the reward signal's properties within each mini-batch. For an input $x$, it generates a group of outputs with advantage values $\{A_i\}_{i=1}^N$. We capture the signal's *magnitude* with the mean of their absolute values, $\mu(x) = \mathbb{E}[|A_i|]$, and its *volatility* with the standard deviation, $\sigma(x)$. The threshold is defined as $\delta(x) = \frac{1}{\beta}\left(\mu(x) + \sigma(x)\right)$. This allows for greater exploration (a higher KL tolerance) when the reward signal is strong or volatile, while tightening the policy stability constraint when the signal is weak and consistent.

With this adaptive threshold, the MKS mechanism manages high-risk samples via an experience buffer ($B$) through a two-stage process:

1. **Identification and Isolation:** If a sample $x$'s KL divergence exceeds the current threshold ($D_{\text{KL}}(\pi_\theta(\cdot|x)\|\pi_{\text{ref}}(\cdot|x)) > \delta(x)$), it is deemed high-risk. Its gradient $\nabla_\theta \mathcal{J}(x)$ is masked, and the sample is moved into the buffer: $B \leftarrow B \cup \{x\}$.

2. **Periodic Re-evaluation:** At a set interval $T$ (e.g., $T = 100$ steps), all samples $x_b$ in the buffer are re-evaluated using the latest policy $\pi_\theta$. Samples whose KL divergence no longer exceeds the new threshold $\delta_t$ are released back into the main training data stream, while the rest remain in the buffer. The buffer is updated as follows:

$$B \leftarrow \{x_b \in B \mid D_{\text{KL}}(\pi_\theta(\cdot|x_b)\|\pi_{\text{ref}}(\cdot|x_b)) > \delta(x_b)\}. \tag{5}$$

This mechanism avoids training collapse by deferring learning from samples currently beyond the model's capabilities, while the periodic re-evaluation ensures valuable difficult samples are re-utilized once the model has improved, thus striking a balance between training stability and data utilization.

## 3.4 ADS: Facilitating Learning Consistency with Adaptive Difficulty Sampling

While MKS provides a reactive safeguard against optimization instability, a more fundamental approach involves proactive regulation of the training data itself. This addresses the core challenge of *inefficient learning*, where a suboptimal data stream leads to slow convergence or unstable training. Existing data scheduling strategies, however, often prove inadequate for complex, long-chain reasoning tasks. Two prevalent paradigms are Curriculum Learning (CL) Parashar et al. (2025); Liu et al. (2024a) and Hard Example Mining (HEM) Zhou et al. (2024). CL typically presents samples in an easy-to-hard progression. While intuitive, this monotonic approach becomes computationally inefficient once the model masters easier concepts, leading to diminishing returns from low-information-gain samples. Conversely, HEM concentrates on samples with the highest error rates. This strategy risks destabilizing the training process, as the "hardest" samples at any given stage may be noisy, adversarial, or simply too far beyond the model's current capabilities. Forcing the model to fit these samples can introduce high-variance gradients, exacerbating the very gradient imbalance that MKS is designed to mitigate.

To address these limitations, we propose **Adaptive Difficulty Sampling (ADS)**, a dynamic, non-monotonic data scheduler inspired by the Zone of Proximal Development McLeod (2012). The core of ADS is a mechanism for dynamically assessing sample difficulty and enforcing learning consistency by continuously aligning the difficulty of the training data with the model's evolving state. We ground our approach in the probability integral transform and implement this by constructing an Empirical Cumulative Distribution Function (ECDF) from the accuracy rewards ($R_{\mathrm{acc}}$) of all samples at the end of each training epoch. This transformation provides a robust, normalized basis for relative difficulty. By applying the ECDF to an individual sample's reward, $\hat{R}_{\mathrm{acc}}^{(i)}$, we obtain its normalized percentile rank, $d(i) = \mathrm{ECDF}(\hat{R}_{\mathrm{acc}}^{(i)})$. This score $d(i) \in [0, 1]$ quantifies the sample's difficulty *relative to all other samples* in the current epoch.

Initially, sampling probabilities $P(i)$ are uniform across the dataset. After each epoch, we use the normalized difficulty scores $d(i)$ to apply a non-monotonic update rule, adjusting the probabilities for the subsequent epoch based on fixed percentile thresholds $\tau_L$ and $\tau_H$ (e.g., 0.2 and 0.8):

$$P_{\mathrm{new}}(i) \propto \begin{cases} \alpha_{\mathrm{easy}} \cdot P_{\mathrm{old}}(i), & \text{if } d(i) > \tau_H \text{ (easy, backprop enabled)} \\ \alpha_{\mathrm{moderate}} \cdot P_{\mathrm{old}}(i), & \text{if } \tau_L \le d(i) \le \tau_H \text{ (moderate, backprop enabled)} \\ \alpha_{\mathrm{hard}} \cdot P_{\mathrm{old}}(i), & \text{if } d(i) < \tau_L \text{ (hard, backprop disabled)} \end{cases} \tag{6}$$

The update coefficients ($\alpha_{\mathrm{easy}} = 0.1, \alpha_{\mathrm{hard}} = 0.8, \alpha_{\mathrm{moderate}} = 1.5$ in our experiments) prioritize moderate-difficulty samples. The resulting unnormalized probabilities are then normalized across the entire dataset to form a valid probability distribution for the next epoch.

The design of ADS directly overcomes the limitations of CL and HEM. By down-weighting and temporarily disabling backpropagation for the hardest samples ($d(i) < \tau_L$), ADS implements a "postponed learning" strategy that avoids the instability of HEM. By reducing the sampling probability of the easiest samples ($d(i) > \tau_H$), it prevents the computational waste of CL, while retaining them at a low probability provides an implicit rehearsal mechanism that mitigates catastrophic forgetting.

## 4 Experiments

### 4.1 Setup

**LVLMs**. Qwen 2.5-VL-3B and InternVL2-2B serve as our base model, selected for its promising capabilities in vision-language understanding, which we aim to further enhance using reinforcement learning. We use the GRPO algorithm based on the VLM-R1 framework Shen et al. (2025) as our main RL baseline, which is designed to enhance the visual reasoning capabilities of LVLMs.

**Training datasets for REC.** To evaluate the generalization of foundational REC skills to advanced reasoning, our model trains on the RefCOCO/+/g splits Mao et al. (2016); Yu et al. (2016), which focus on visual attributes like object location and appearance rather than multi-step or abstract reasoning. **Evaluation datasets for REC.** ID performance is measured on the validation and test splits of RefCOCO/+/g Mao et al. (2016); Yu et al. (2016). For OOD generalization, we use RefGTA Tanaka et al. (2019) to test visual domain shift with synthetic human images, and the LISA-Grounding

Table 1: A comprehensive performance comparison on visual grounding benchmarks. The table contrasts general foundation models with results on Qwen2.5VL-3B and InternVL-2B base models.

| Model | RefCOCO | | | RefCOCO+ | | | RefCOCOg | | Avg. |
|---|---|---|---|---|---|---|---|---|---|
| | val | testA | testB | val | testA | testB | val | test | |
| *Part 1: General State-of-the-Art Models* | | | | | | | | | |
| Grounding DINO-Tiny Liu et al. (2023b) | 89.2 | 91.9 | 86.0 | 81.1 | 87.4 | 74.7 | 85.2 | 84.9 | 85.1 |
| Grounding DINO-Large Liu et al. (2023b) | 90.6 | 93.2 | 88.2 | 82.8 | 89.0 | 75.9 | 86.1 | 87.0 | 86.6 |
| HieA2G Wang et al. (2025) | 87.8 | 90.3 | 84.0 | 80.7 | 85.6 | 72.9 | 83.7 | 83.8 | 83.6 |
| InternVL2-1B Team (2024) | 83.6 | 88.7 | 79.8 | 76.0 | 83.6 | 67.7 | 80.2 | 79.9 | 79.9 |
| *Part 2: Comparison on Qwen-2.5VL-3B Base Model* | | | | | | | | | |
| Qwen2.5VL-3B Bai et al. (2025) | 88.4 | 91.2 | 83.6 | 80.6 | 86.9 | 72.8 | 84.2 | 84.7 | 84.1 |
| + GRPO | 90.3(+1.9) | 92.5(+1.3) | 85.7(+2.1) | 84.3(+3.7) | 89.4(+2.5) | 77.1(+4.3) | 86.1(+1.9) | 86.8(+2.1) | 86.5(+2.4) |
| + TACO (Ours) | 91.8(+3.4) | 93.4(+2.2) | 87.7(+4.1) | 86.3(+5.7) | 90.8(+3.9) | 79.7(+6.9) | 87.8(+3.6) | 88.3(+3.6) | 88.2(+4.1) |
| *Part 3: Comparison on InternVL2-2B Base Model* | | | | | | | | | |
| InternVL2-2B Team (2024) | 82.3 | 88.2 | 75.9 | 73.5 | 82.8 | 63.3 | 77.6 | 78.3 | 77.7 |
| + GRPO | 83.7(+1.4) | 89.2(+1.0) | 77.2(+1.3) | 75.3(+1.8) | 83.9(+1.1) | 65.4(+2.1) | 78.9(+1.3) | 79.2(+0.9) | 79.1(+1.4) |
| + TACO (Ours) | 84.6(+2.3) | 90.1(+1.9) | 78.6(+2.7) | 76.9(+3.4) | 84.9(+2.1) | 67.2(+3.9) | 80.4(+2.8) | 82.3(+2.1) | 80.6(+2.9) |

Table 2: Performance comparison on OOD LISA-Grounding Benchmark.

| Model | LISA | RefGTA |
|---|---|---|
| Qwen2.5VL-3B Bai et al. (2025) | 55.4 | 70.8 |
| + GRPO | 61.1 | 71.6 |
| + TACO (Ours) | 75.1 | 78.7 |
| InternVL-2B Team (2024) | 48.6 | 63.3 |
| + GRPO | 51.9 | 64.1 |
| + TACO (Ours) | 62.1 | 68.3 |

Table 3: Performance comparison on OCR-related Understanding Tasks.

| Model | InfoVQA | TextVQA | DocVQA |
|---|---|---|---|
| | (VAL) | (VAL) | (VAL) |
| Qwen2.5VL-3B Bai et al. (2025) | 75.1 | 78.7 | 93.0 |
| + GRPO | 76.1 | 78.4 | 92.7 |
| + TACO (Ours) | 77.7 | 79.7 | 93.2 |
| InternVL-2B Team (2024) | 58.9 | 73.4 | 86.9 |
| + GRPO | 59.1 | 73.6 | 86.8 |
| + TACO (Ours) | 61.2 | 74.4 | 87.3 |

test split Lai et al. (2024) to assess reasoning transfer in tasks requiring fine-grained visual-linguistic and relational understanding.

**Training datasets for VQA.** For VQA training, we compiled a dataset of 9,600 instances by randomly sampling from multiple sub-datasets in the R1-Vision collection Shen et al. (2024), including MathQA, ChartQA, DeepForm, DocVQA, InfographicsVQA, TextVQA, and OCRVQA. This sample size aligns with our 800-step training procedure. **Evaluation datasets for VQA.** VQA performance is evaluated using specialized datasets such as MMStar Chen et al. (2024a), AI2D Kembhavi et al. (2016), InfoVQA VAL Mathew et al. (2022), TextVQA VAL Singh et al. (2019), DocVQA VAL Mathew et al. (2021), MATH-Vision-FULL Wang et al. (2024), and MMBench Liu et al. (2024b), testing the model's capabilities across various VQA tasks.

**Training datasets for Video VQA.** Long-Horizon Video VQA benchmark comprises 4,000 question-answer samples paired with real-world videos of 20-60 seconds in duration, which is designed to evaluate a model's robustness in maintaining complex reasoning logic and policy stability over extended, dynamic visual inputs.

## 4.2 EXPERIMENTAL RESULTS

**Comparison to State of the Art REC.** TACO demonstrates state-of-the-art performance and superior learning potential across various REC benchmarks. First, in a head-to-head comparison of final performance against existing SOTA models (see Table 1), TACO achieves the best results on all test splits of RefCOCO/+/g. It not only significantly surpasses the baseline GRPO algorithm but also outperforms specialized models like Grounding DINO. This initially validates the high performance ceiling that the TACO framework can achieve upon convergence.

To delve deeper into the origins of this performance advantage, we analyze the **longitudinal learning dynamics** in Table 4. This table reveals the learning capacity and generalization potential of different methods. On in-domain (ID) datasets, the performance of SFT rapidly saturates after 200 steps, while the baseline GRPO, though continuously learning, shows limited gains. In contrast, TACO exhibits the steepest and most sustained learning curve, demonstrating that **ADS** effectively prevents learning stagnation by continuously supplying high-information-gain signals through efficient data scheduling.

Table 4: Performance (accuracy) comparison of SFT and RL methods on ID and OOD benchmarks. Scores for RefCOCO/+/g represent average accuracies across sub-datasets (see Appendix for details). All models are based on Qwen 2.5-VL-3B, with SFT and RL using RefCOCO/+/g training splits. Scores at "Step 0" correspond to the Qwen 2.5-VL-3B model. $\Delta_{RL-SFT}$ represents the RL model's gain over SFT, and $\Delta_{RL-GRPO}$ shows our model's advantage over GRPO.

| Training Method | Evaluation Dataset | Training Steps | | | | |
|---|---|---|---|---|---|---|
| | | 0 | 200 | 400 | 600 | 800 |
| SFT | | 87.79 | 88.08 | 88.16 | 88.21 | 88.27 |
| GRPO | Refcoco | 87.79 | 88.80 | 89.12 | 89.30 | 89.42 |
| Ours | | 87.79 | 89.55 | 89.96 | 90.36 | 90.42 |
| SFT | | 80.63 | 81.60 | 81.31 | 81.19 | 81.28 |
| GRPO | Refcoco+ | 80.63 | 82.39 | 82.64 | 83.26 | 83.50 |
| Ours | | 80.63 | 83.45 | 84.32 | 84.67 | 85.00 |
| SFT | | 84.79 | 85.02 | 84.80 | 84.59 | 84.68 |
| GRPO | Refcocog | 84.79 | 85.36 | 85.86 | 86.38 | 86.46 |
| Ours | | 84.79 | 86.57 | 87.31 | 87.40 | 87.70 |
| SFT | | 55.37 | 56.15 | 54.95 | 54.16 | 54.83 |
| GRPO | LISA-Grounding | 55.37 | 61.76 | 62.00 | 60.68 | 61.10 |
| Ours | | 55.37 | 62.97 | 64.49 | 65.26 | 66.44 |
| $\Delta_{Ours-SFT}$ | | 0 | +6.82 | +9.54 | +11.10 | +11.61 |
| $\Delta_{Ours-GRPO}$ | | 0 | +1.21 | +2.49 | +4.58 | +5.34 |

Table 5: Performance comparison on diverse multimodal benchmarks using Qwen2.5VL-3B and InternVL-2B base models.

| Model | Math Vision | MMBench | | | | MMStar | AI2D |
|---|---|---|---|---|---|---|---|
| | (Full) | EN(dev) | CN(dev) | EN-V11 | CN-V11 | (test) | (test) |
| Qwen2.5VL-3B Bai et al. (2025) | 20.1 | 78.0 | 77.2 | 75.8 | 75.6 | 53.0 | 77.4 |
| + GRPO | 21.2 | 79.7 | 77.9 | 78.0 | 76.7 | 54.4 | 78.4 |
| **+ TACO (Ours)** | **24.1** | **81.1** | **79.0** | **79.3** | **78.0** | **59.9** | **80.5** |
| InternVL-2B Team (2024) | – | 70.4 | 71.1 | 69.8 | 67.6 | 50.0 | 74.1 |
| + GRPO | – | 71.5 | 72.2 | 70.9 | 68.7 | 50.9 | 75.2 |
| **+ TACO (Ours)** | – | **72.8** | **73.3** | **72.6** | **69.4** | **52.1** | **76.7** |

This advantage is drastically amplified on the OOD LISA-Grounding benchmark, which demands complex long-chain reasoning. The SFT method fails entirely on this task, proving its inability to generalize. The baseline GRPO, while superior to SFT, also stagnates after 600 steps, empirically validating our thesis on optimization inconsistency—unconstrained exploration eventually hits a bottleneck caused by gradient conflicts. TACO, however, demonstrates sustained and stable performance growth, achieving a performance gain of **+5.34%** over GRPO ($\Delta_{Ours-GRPO}$) at 800 steps. This provides evidence of the critical synergy between the long-chain exploration incentivized by **TAC** and the stability provided by **MKS**, which allows the model to convert high-risk exploration into generalizable reasoning abilities. As further confirmed in Table 2, TACO's performance on both LISA and RefGTA consistently surpasses all baseline methods.

**VQA.** The superiority of TACO also extends to a wide array of VQA tasks. As shown in Tables 5 and 3, TACO consistently outperforms the strong Qwen2.5VL-3B base model across multiple benchmarks, including general VQA, math reasoning, chart understanding, and OCR-related tasks. The significant improvement of **+6.9%** on the challenging MMStar benchmark is particularly noteworthy. This indicates that the TACO framework enhances a general, fundamental complex reasoning ability rather than task-specific skills. By systematically enforcing consistency at the **semantic**, **optimization**, and **learning** levels, TACO enables the model to more efficiently learn generalizable and compositional knowledge, leading to superior performance across a diverse set of challenges.

**Long-Horizon Video VQA.** To further test TACO's stability on long-horizon reasoning tasks, we evaluated it on a real-world video VQA benchmark. This task involves queries that require intricate, step-by-step analysis across video frames (20-60 seconds), posing a significant challenge to policy stability. The results, presented in Table 6, show that TACO (94.80%) substantially outperforms the baseline GRPO (81.34%), which falters on such long-horizon tasks. This provides decisive evidence that TACO's MKS and ADS components are critical for mitigating exploration collapse and maintaining learning consistency, ensuring stable and effective reasoning over extended sequences.

Table 6: Performance on the long-horizon Video VQA benchmark.

| Method | Accuracy (%) |
|---|---|
| Qwen-2.5VL-7B | 69.50 |
| + GRPO | 81.34 |
| **+ TACO (Ours)** | **94.80** |

Table 7: Ablation results of our method on different datasets, evaluating the individual contributions of each component.

| TAC | RRS | ADS | RefCOCO | RefCOCO+ | RefCOCOg | LISA |
|---|---|---|---|---|---|---|
| | | | 89.5 | 83.6 | 86.5 | 61.2 |
| ✓ | | | 90.1 | 84.6 | 87.1 | 70.4 |
| ✓ | ✓ | | 90.5 | 85.1 | 87.4 | 72.9 |
| ✓ | ✓ | ✓ | **90.8** | **85.6** | **87.6** | **75.1** |

## 4.3 ABLATION STUDIES

**Component Analysis** We conducted a series of ablation studies to systematically dissect the individual and collective contributions of the TACO framework's core components: Think-Answer Consistency (TAC), Memory-Guided KL Stabilization (MKS), and Adaptive Difficulty Sampling (ADS). The results, summarized in Table 7, progressively build upon the GRPO baseline to isolate the impact of each mechanism. The results show that introducing TAC alone significantly improves performance by enforcing semantic consistency, confirming its role in correcting ineffective reasoning. Layering on MKS further enhances results, demonstrating its necessity for stabilizing the long-chain exploration that TAC encourages, thereby preventing optimization collapse. The complete framework with ADS achieves the best performance by maximizing learning efficiency via adaptive data scheduling. These results confirm that the components are indispensable and synergistic: TAC provides a valid objective, MKS ensures stable optimization, and ADS accelerates learning.

**Comparison of MKS and ADS** The MKS and ADS mechanisms, while functionally distinct, are designed to be complementary, addressing different facets of the training process. MKS serves as a reactive safeguard for optimization stability, while ADS acts as a proactive regulator of learning efficiency by scheduling data based on reward signals. An experiment on a 1,200-sample subset, summarized in Table 8, validates this synergy. Initially, 298 samples were identified as "high-risk" and isolated by MKS due to excessive KL divergence. As the model's policy improved on the ADS-curated data stream, 266 (89%) of these were later successfully re-integrated into the training process. This

Table 8: Synergy between MKS and ADS on a 1,200-sample subset. It details the initial sample distribution and the transitions during training, highlighting the interplay between them.

| Metric | Count |
|---|---|
| *Initial Distribution* | |
| MKS (High-Risk) | 298 |
| ADS (Learnable) | 902 |
| *Transitions During Training* | |
| MKS → ADS (Became Learnable) | 266 |
| ADS → MKS (Became High-Risk) | 0 |

demonstrates that MKS implements an effective "postponed learning" strategy, preserving valuable data until the model can learn from them without collapsing. Critically, the number of samples transitioning from the ADS "learnable" pool into the MKS "high-risk" buffer was zero. This finding underscores that the data stream curated by ADS is inherently stable, preemptively filtering out inputs likely to cause instability.

## 5 CONCLUSION

This work addresses the systemic challenge of maintaining consistency across semantic, optimization, and learning levels during the reinforcement learning of Large Vision-Language Models (LVLMs). We identify that common failure modes like ineffective reasoning, exploration collapse, and inefficient learning are manifestations of this underlying multi-level inconsistency. To tackle this, we introduce TACO, a unified framework where three synergistic mechanisms—Think-Answer Consistency (TAC), Memory-Guided KL Stabilization (MKS), and Adaptive Difficulty Sampling (ADS)—work in concert to enforce semantic, optimization, and learning consistency, respectively. This integrated design enables TACO to achieve state-of-the-art performance across 15 REC and VQA benchmarks while demonstrating superior generalization on complex reasoning tasks where conventional methods falter. The effectiveness of TACO underscores a fundamental insight: unlocking the full potential of RL-based training for complex reasoning requires a holistic approach that ensures coherence across all dimensions of the learning process.

## ETHICS STATEMENT

This research was conducted in alignment with the ICLR Code of Ethics. Our work is based exclusively on publicly available datasets, all of which were used in a manner that respects their intended use licenses and terms. These datasets include RefCOCO/+/g, RefGTA, LISA-Grounding, the R1-Vision collection (from which we sampled MathQA, ChartQA, DeepForm, DocVQA, InfographicsVQA, TextVQA, and OCRVQA), MMStar, AI2D, MATH-Vision-FULL, MMBench, and the Long-Horizon Video VQA benchmark. The study did not involve human subjects or animal experimentation, and no personally identifiable information was processed. We were conscious of and took steps to prevent potential biases in our data handling and model evaluation processes. We are committed to responsible research practices and transparency.

## REPRODUCIBILITY STATEMENT

To ensure our results can be fully reproduced, we have included our source code with detailed annotations in the appendix. Furthermore, we commit to open-sourcing the entire codebase and all experimental configurations upon publication. The main paper provides a thorough account of our experimental setup, including training procedures and model hyperparameters. All datasets used for training and evaluation, such as RefCOCO/+/g, LISA-Grounding, MMBench, and MMStar, are public benchmarks, which enables consistent and comparable evaluation.

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

# A    LLM USAGE

In preparing this manuscript, we utilized a Large Language Model (LLM) as a writing support tool. Its application was strictly limited to stylistic improvements, such as enhancing sentence structure, improving readability, and checking for grammatical correctness. All scientific contributions, including the core concepts, experimental design, reinforcement learning methodology, and analysis of the results, originate solely from the authors. The LLM had no role in generating any of the substantive, scientific content of this paper. The authors take full and final responsibility for all information and claims presented herein.

# B    ADDITIONAL EXPERIMENTAL RESULTS

Table 9: Performance comparison of SFT and RL on in-domain and out-of-domain evaluation datasets. All results are from Qwen 2.5-VL-3B trained on the training split of Refcoco/+/g. Step 0 represents the results from Qwen 2.5-VL-3B itself. $\Delta_{RL-SFT}$ denotes the improved value of the RL model compared to the SFT model. $\Delta_{RL-GRPO}$ denotes the improved value of ours compared to GRPO.

| Training Method | Evaluation Dataset | Training Steps | | | | |
|---|---|---|---|---|---|---|
| | | 0 | 200 | 400 | 600 | 800 |
| SFT | | 88.19 | 88.78 | 88.90 | 88.99 | 88.94 |
| GRPO | Refcoco$_{val}$ | 88.19 | 89.72 | 89.79 | 90.19 | 90.28 |
| Ours | | 88.19 | 90.55 | 91.10 | 91.28 | 91.40 |
| SFT | | 91.51 | 92.13 | 92.20 | 91.97 | 92.13 |
| GRPO | Refcoco$_{test_A}$ | 91.51 | 92.12 | 92.08 | 92.65 | 92.63 |
| Ours | | 91.51 | 92.63 | 93.02 | 93.37 | 93.44 |
| SFT | | 83.67 | 83.32 | 83.38 | 83.67 | 83.75 |
| GRPO | Refcoco$_{test_B}$ | 83.67 | 84.55 | 85.48 | 85.06 | 85.36 |
| Ours | | 83.67 | 86.24 | 86.42 | 86.93 | 87.09 |
| SFT | | 81.08 | 82.15 | 81.93 | 82.01 | 82.01 |
| GRPO | Refcoco+$_{val}$ | 81.08 | 83.16 | 83.60 | 83.96 | 84.29 |
| Ours | | 81.08 | 84.33 | 85.16 | 85.36 | 85.58 |
| SFT | | 87.37 | 88.56 | 88.32 | 88.25 | 88.09 |
| GRPO | Refcoco+$_{test_A}$ | 87.37 | 88.35 | 88.42 | 89.45 | 89.38 |
| Ours | | 87.37 | 89.37 | 89.84 | 90.03 | 90.25 |
| SFT | | 73.43 | 74.08 | 73.68 | 73.31 | 73.74 |
| GRPO | Refcoco+$_{test_B}$ | 73.43 | 75.66 | 75.91 | 76.38 | 76.83 |
| Ours | | 73.43 | 76.72 | 77.95 | 78.63 | 79.16 |
| SFT | | 84.56 | 85.05 | 84.72 | 84.33 | 84.48 |
| GRPO | Refcocog$_{val}$ | 84.56 | 85.27 | 85.64 | 86.32 | 86.17 |
| Ours | | 84.56 | 86.30 | 87.01 | 87.40 | 87.62 |
| SFT | | 85.02 | 84.98 | 84.87 | 84.84 | 84.87 |
| GRPO | Refcocog$_{test}$ | 85.02 | 85.44 | 86.08 | 86.43 | 86.74 |
| Ours | | 85.02 | 86.83 | 87.60 | 87.40 | 87.77 |
| SFT | | 55.37 | 56.15 | 54.95 | 54.16 | 54.83 |
| GRPO | | 55.37 | 61.76 | 62.00 | 60.68 | 61.10 |
| Ours | LISA-Grounding | 55.37 | 61.94 | 63.69 | 63.26 | 65.50 |
| $\Delta_{Ours-SFT}$ | | 0 | +5.70 | +8.70 | +9.10 | +10.70 |
| $\Delta_{Ours-GRPO}$ | | 0 | +0.10 | +1.70 | +2.60 | +4.40 |

## Prompt Template

### REC

"{Question} First output the thinking process then summarize the answer in <think> </think> tags and then output the final answer in <answer> </answer> tags. Output the final answer in JSON format."

### VQA

"{Question} First output the thinking process in <think> </think> tags and then output the final answer in <answer> </answer> tags. Output the final answer in JSON format."

#### Consistency judgment prompt in VQA

```
messages=[
            {
                "role": "user",
                "content": "As a text comprehension expert, evaluate
        the semantic consistency between automatically extracted answers
        based on the given corpus, questions, and reference answers,
        outputting a similarity score within the [0,1] range. Output ONLY
        score"
            },
            {
                "role": "user",
                "content": f"given corpus:{think_content}, reference
        answers: {solution}\nquestions: {problem}\nOutput score only"
            }
        ]
```

Figure 4: Prompt templates for REC, VQA, and VQA consistency judgment, illustrating specific instructions for thought process elicitation, JSON answer formatting, and semantic consistency evaluation.

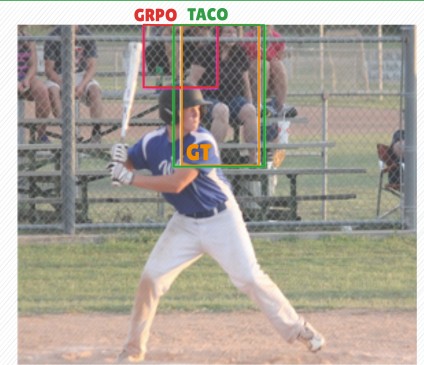

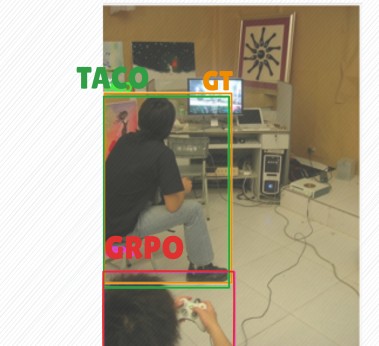

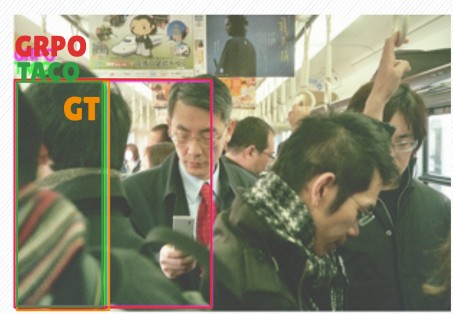

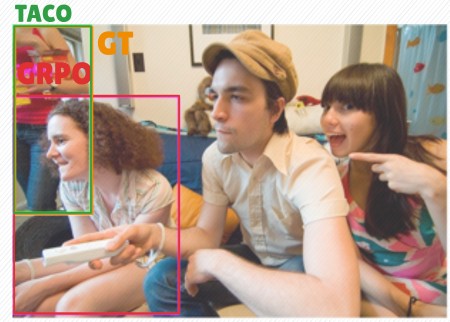

Figure 5: In-domain case studies from RefCOCO/+/g test splits, comparing GRPO with our method. Both models use Qwen 2.5-VL-3B and were trained on RefCOCO/+/g. Bounding boxes: GT (orange), GRPO (red), Our Method (green).

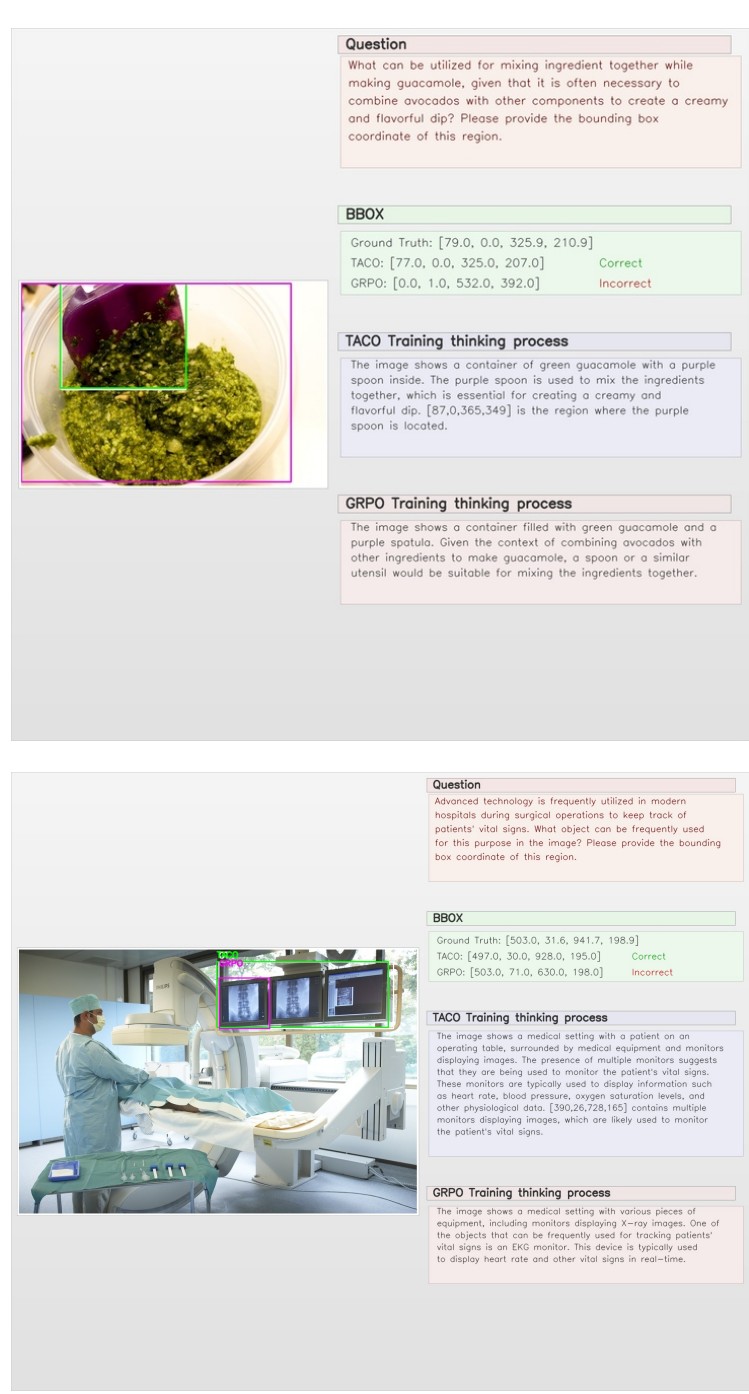

Figure 6: Qualitative examples of visual grounding on LISA dataset. TACO (green), GRPO (purple), ground truth (white).

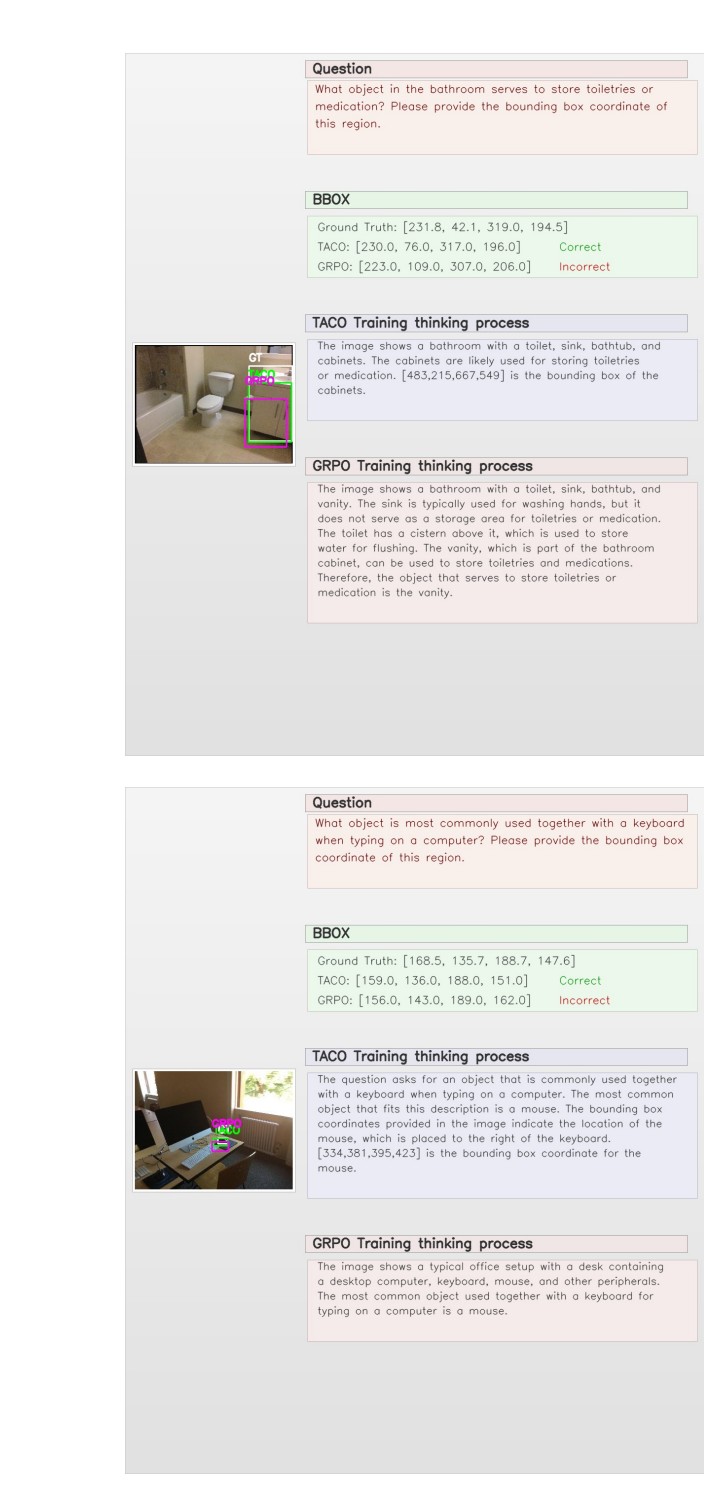

Figure 7: Qualitative examples of visual grounding on LISA dataset. TACO (green), GRPO (purple), ground truth (white).

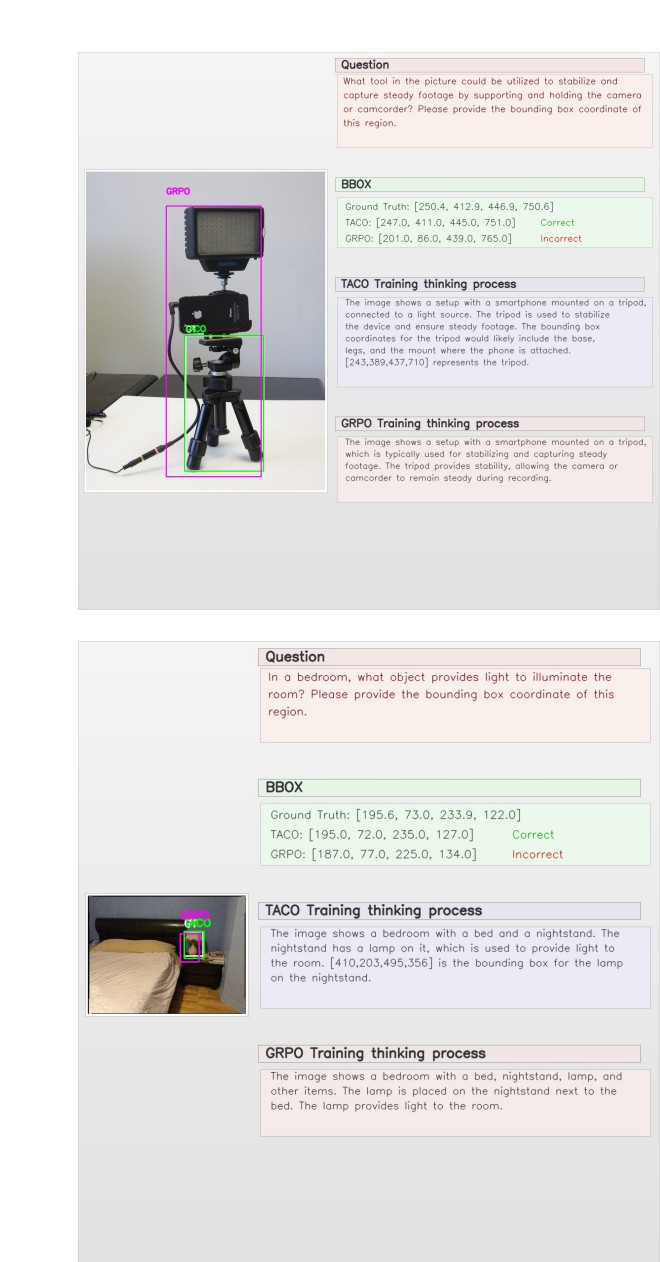

Figure 8: Qualitative examples of visual grounding on LISA dataset. TACO (green), GRPO (purple), ground truth (white).

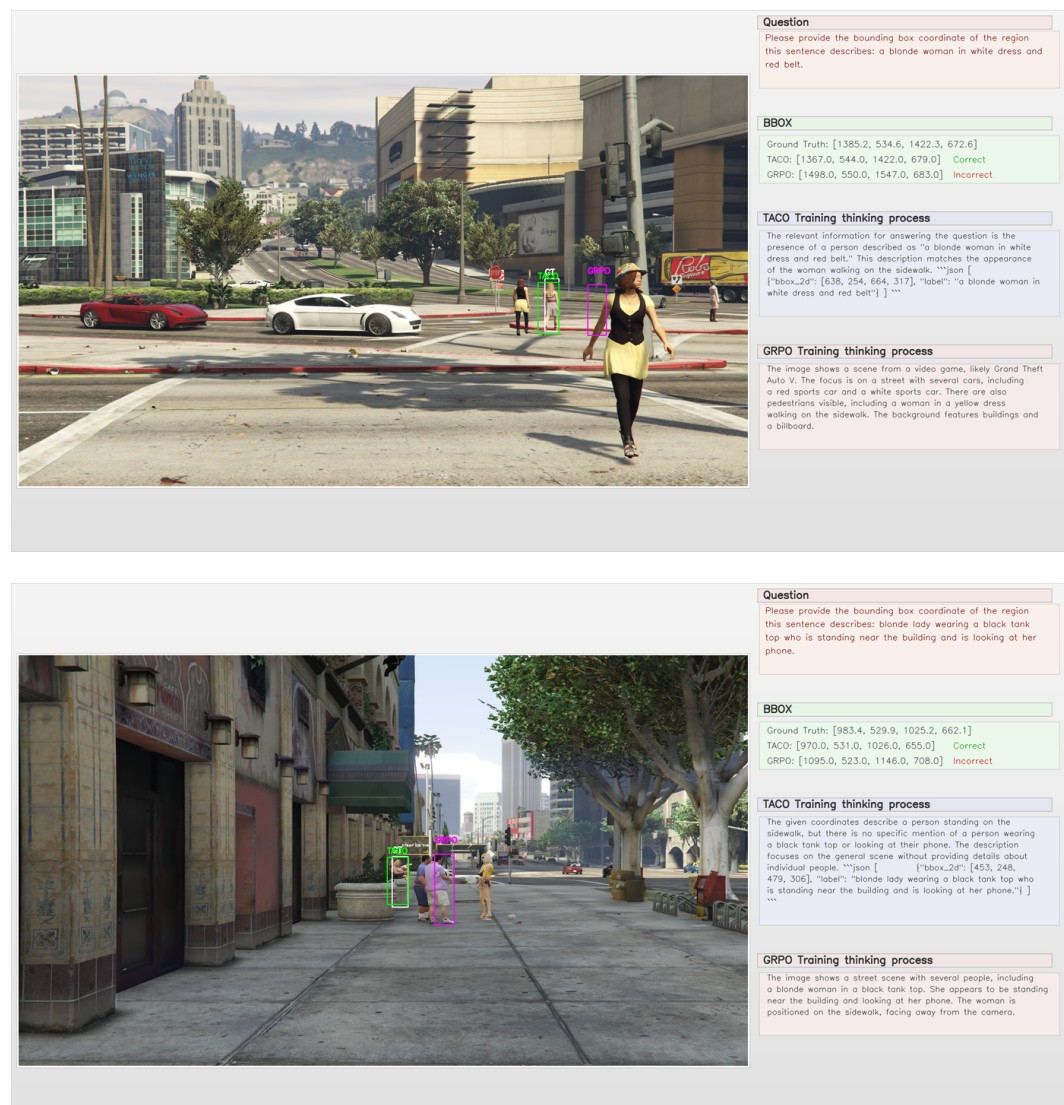

Figure 9: Qualitative examples of visual grounding on RefGTA dataset. TACO (green), GRPO (purple), ground truth (white).

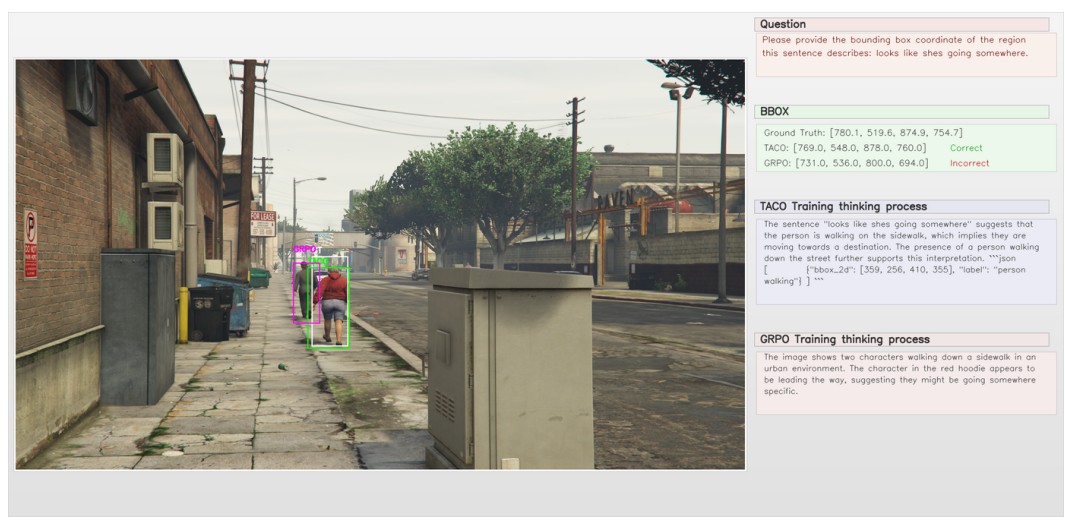

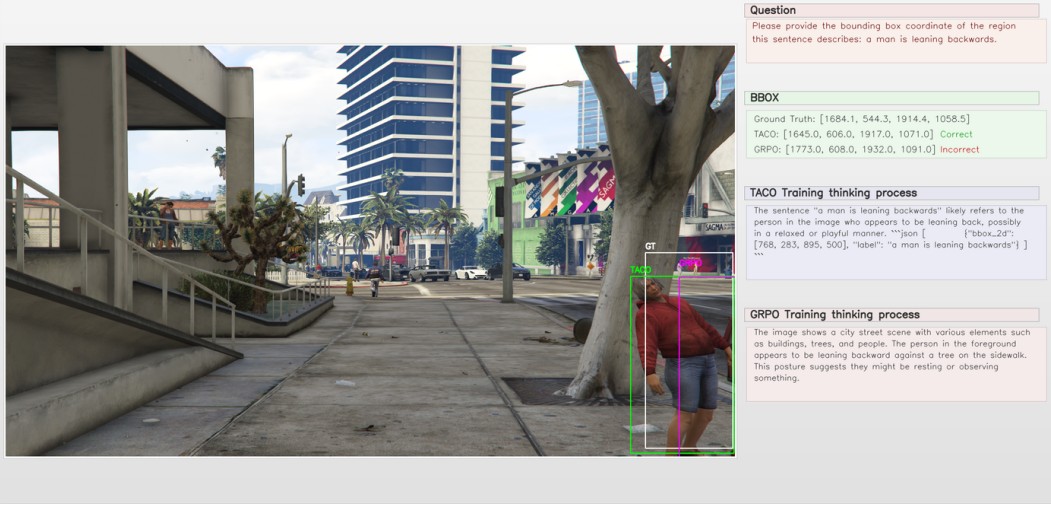

Figure 10: Qualitative examples of visual grounding on RefGTA dataset. TACO (green), GRPO (purple), ground truth (white).

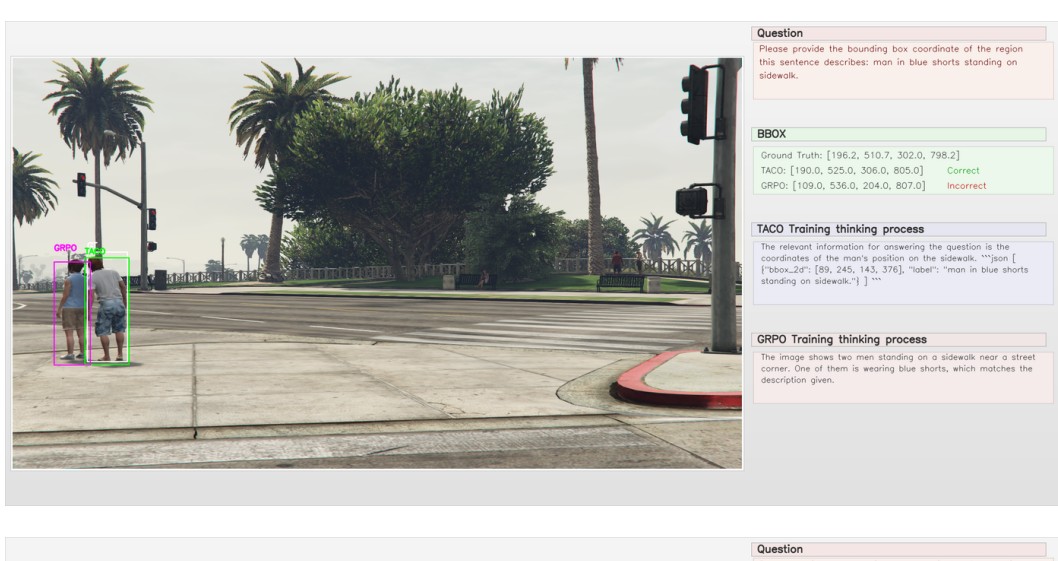

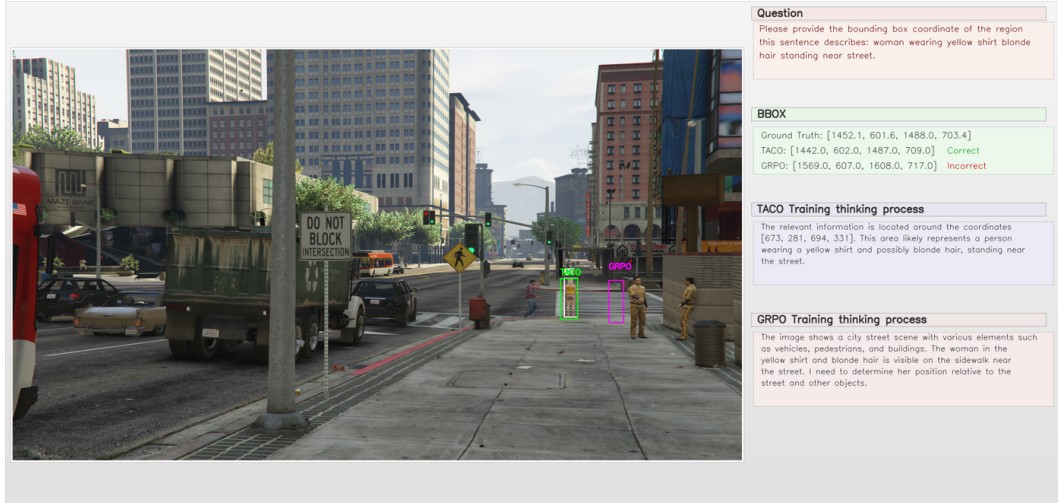

Figure 11: Qualitative examples of visual grounding on RefGTA dataset. TACO (green), GRPO (purple), ground truth (white).

# Statistical Reasoning

| Case | Amount of annuity | Interest rate | Deposit period (years) |
|------|-------------------|---------------|------------------------|
| A | $ 2,500 | 8% | 10 |
| B | 500 | 12 | 6 |
| C | 30,000 | 20 | 5 |
| D | 11,500 | 9 | 8 |
| E | 6,000 | 14 | 30 |

Question: For case A accompanying table, answer the questions that follow.<image 1>Calculate the future value of the annuity, assuming that it is an ordinary annuity.

-

A. $126,827
B. $223,248
C. $36,216
D. $4,058
Please select the correct answer from the options above.

**TACO**

```
<think>The image shows a table with five cases, each having different
parameters such as the amount of annuity, interest rate, and deposit period. We
need to calculate the future value of the annuity for Case A, which has an
amount of $2,500, an interest rate of 8%, and a deposit period of 10 years.

To solve this problem, we use the formula for the future value of an ordinary
annuity:

\[ FV=P \times \frac{(1+r)^n - 1}{r} \]

Where :
- \(P \)is the payment amount ($2,500),
- \(r \)is the interest rate (8% or 0.08),
- \(n \)is the number of periods (10 years).

Plugging in the values:

\[ FV = 2500 \times \frac{(1+0.08)^{10} - 1}{0.08} \]

Calculating step by step:
1. Calculate \((1 + 0.08)^{10} \): \[(1.08)^{10} \approx 2.15892 \]
2. Subtract 1:\[2.15892 - 1 = 1.15892 \]
3. Divide by the interest rate:
   \[ \frac{1.15892}{0.08} \approx 14.4865\]
4. Multiply by the payment amount:
   \[2500 \times 14.4865 \approx 36,216 \]

Therefore, the future value of the annuity for Case A is approximately $36,216.
Looking at the options provided, C is $36,216, so that must be the correct
answer.(candidate)C(/candidate)</think><answer>{\"answer\": \"C\"</answer>
```

**Base Model**

```
B. $223,248
```

Figure 12: Case studies on the VQA training dataset and tests on statistical reasoning tasks were conducted. We compare the response generated by our method and Qwen 2.5-VL-3B base model. The response given by the base model is incorrect, while our method (TACO) is correct.

# Math

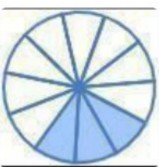

Question: Hint: Please answer the question and provide the correct option letter, e.g., A, B, C, D, at the end.
Question: What fraction of the shape is blue?
Options:
A. 3/11
B. 8/11
C. 6/11
D. 3/5
Please select the correct answer from the options above.

## TACO

```
<think>The image shows a circle divided into 11 equal
segments by lines radiating outward. Out of these 11
segments, three are shaded blue. Therefore, the fraction
of the shape that is blue is calculated as the number of
blue segments divided by the total number of segments.

So,the fraction is 3/11.

Looking at the options provided:

A.3/11
B.8/11
C.6/11
D.3/5

The correct answer is A.(candidate)A(/candidate)</think>
<answer>{\"answer\":\"A\"}</answer>
```

## Base Model

```
C. 6/11
```
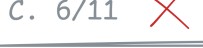

Figure 13: Case studies on VQA training dataset and tests on mathematical tasks were conducted. We compare the responses generated by our method and Qwen 2.5-VL-3B base model. The response given by the base model is incorrect, while our method (TACO) is correct.

# Chart Q&A

| Proprietary | | | | |
|---|---|---|---|---|
| GPT-4O | N.A. | - | - | 85.7 |
| GPT-4V | N.A. | - | - | 78.5 |
| Public Large Models | | | | |
| LLaVA OneVision-72B | 72B | - | - | 83.7 |
| Llama 3-V 70B | >70B | - | - | 83.2 |
| Public Small Models | | | | |
| Llama 3-V 8B | >8B | - | - | 78.7 |
| ChartLlama | 7B | 48.96 | 90.36 | 69.7 |
| LLaVA OneVision-7B | 7B | - | - | 80.0 |
| SynChart | 4.2B | 74.24 | 94.96 | 84.6 |

### 4.2 Ablations

## TACO

The image provided is a table that summarizes the performance of various
large language models across different categories (proprietary,public large
models, and public small models). The table includes the model names, their
sizes, and corresponding scores for two tasks: "Ablations" and "Other."

Let\'s summarize the key points from the table:

1. **Propriety Models**:
- GPT-40: N.A.
- GPT-4V: N.A.

2. **Public Large Models**:
- LLCVA OneVision-72B: Score: 83.7
- Llama 3-V 70B: >70B, Score: 83.2

3. **Public Small Models**:
- Llama 3-V 8B: >8B, Score: 78.7
- ChartLlama: 7B, Score:90.36
- LlaVA OneVision-7B: 7B, Score: 80.0
- SynChart: 4.2B, Score: 94.96

- From the scores listed under each model, we can see the highest score
among all models is 94.96, which is associated with the **Sync Chart**
model.

Analyzing the scores:
- The highest score among all models is 94.96, which is achieved by the
 **SynChart** model.

(candidate)SynChart(/candidate)</think><answer>{"answer":"SynChart"}
</answer>

## Base Model

According to the data in the table, the Synchart model performs the best on all
evaluation metrics. Specifically\n\nSynchart scored 84.6 in the Proprietary
evaluation.\n\nSynchart scored 83.2 in the Public Large Models evaluation.\n\nSynchart
scored 94.96 in the Public Small Models evaluation.\n\nTherefore, Synchart is the best-
performing model on these three evaluation metrics.

Figure 14: Case studies on VQA training dataset and tests on Chart-Q&A tasks were conducted. We
compare the responses generated by our method and Qwen 2.5-VL-3B base model. The base model
lacks the thoroughness and logical rigor of our method(TACO).

