# OpenReview forum: "TACO: Think-Answer Consistency for Optimized Long-Chain Reasoning and Efficient Data Learning via Reinforcement Learning in LVLMs"
_ICLR.cc/2026/Conference — ICLR 2026 Conference Withdrawn Submission_

### Official Review · Reviewer_yUCh · 2025-10-18

**Soundness:** 2
**Presentation:** 2
**Contribution:** 3
**Rating:** 4
**Confidence:** 4

**Summary:**

This paper proposes TACO, a reinforcement learning framework for training Large Vision-Language Models (LVLMs) that addresses three identified failure modes: exploration collapse, inefficient learning, and ineffective reasoning. The framework consists of three synergistic components: (1) Think-Answer Consistency (TAC) reward that enforces alignment between reasoning traces, final answers, and ground truth; (2) Memory-Guided KL Stabilization (MKS) that defers high-risk samples to an experience buffer to prevent optimization collapse; and (3) Adaptive Difficulty Sampling (ADS) that dynamically adjusts data sampling based on difficulty. The authors evaluate TACO on 15 benchmarks spanning Referring Expression Comprehension (REC), Visual Question Answering (VQA), and Video VQA tasks, demonstrating improvements over baseline GRPO methods.

**Strengths:**

Well-motivated problem formulation: The paper clearly articulates three distinct failure modes in LVLM training and frames them as consistency failures across semantic, optimization, and learning levels. This unified perspective is valuable.
Comprehensive experimental validation: The evaluation spans diverse tasks (REC, VQA, Video VQA) and includes both in-domain and out-of-domain benchmarks, demonstrating broad applicability.
Thorough ablation studies: Table 7 and Table 8 provide useful ablations showing the individual and synergistic contributions of each component, particularly the interaction between MKS and ADS.
Strong empirical results: The method shows consistent improvements across multiple benchmarks, with particularly impressive gains on challenging tasks like LISA-Grounding (+14.1% over baseline) and long-horizon Video VQA (+13.46% over GRPO).
Analysis of learning dynamics: Table 4's longitudinal analysis effectively demonstrates sustained learning compared to baselines that plateau, supporting the efficiency claims.

**Weaknesses:**

Limited technical novelty: While the combination is novel, the individual components lack significant innovation:

TAC for REC is simply the IoU of three bounding boxes—a straightforward geometric constraint

MKS resembles standard experience replay with adaptive thresholding

ADS is curriculum learning with fixed percentile thresholds

The paper would benefit from clearer articulation of what is technically novel beyond the combination


Circular dependency on external supervisor: For VQA tasks, TAC relies on a much larger external model (Qwen-32B) to evaluate reasoning quality (Eq. 4). This raises critical concerns:

Creates a supervision bottleneck: the student model is limited by the supervisor's capabilities

Adds substantial computational cost (not quantified)

Undermines the claim of "autonomous problem-solving" if a larger model is needed for evaluation

No justification for why this is superior to self-consistency checking or other approaches

The supervisor's own potential biases and errors are not discussed


Insufficient experimental rigor:

No error bars or confidence intervals: Given RL's stochastic nature, single-run results are insufficient

Custom video benchmark: The 4,000-sample Video VQA dataset lacks details on creation, validation, diversity, or public availability, making it impossible to assess quality or reproduce results

Limited baseline comparisons: Only compares against GRPO despite citing multiple recent VLM-RL methods (R1-V, Visual-RFT, VLM-R1) without comparison

Small training sets: 9,600 VQA samples and 4,000 Video samples are quite small—unclear if findings generalize to large-scale training


Missing computational cost analysis: The framework adds significant overhead:

Experience buffer management and periodic re-evaluation (every T=100 steps)

ECDF computation for all samples each epoch

External supervisor model inference for VQA

No wall-clock time, memory usage, or throughput comparisons are provided

**Questions:**

Can you provide computational cost comparisons (wall-clock time, memory, FLOPs) between TACO and baseline GRPO?

For VQA TAC, have you considered self-consistency methods instead of external supervision? What is the performance-cost tradeoff?

Can you provide qualitative examples demonstrating the "ineffective reasoning" problem and how TAC addresses it?

Why is disabling backpropagation for hard samples preferable to alternative strategies like gradient clipping or learning rate scaling?

Have you tested TACO on larger models (7B+)? What are the scaling properties?

Can you provide error bars from multiple runs with different seeds?

Will the video VQA benchmark and all code be publicly released?

---

### Official Review · Reviewer_oBjA · 2025-10-21

**Soundness:** 2
**Presentation:** 2
**Contribution:** 2
**Rating:** 2
**Confidence:** 3

**Summary:**

This work proposes the TACO framework, which aims to ensure the stability of reinforcement learning training from the perspective of data sampling. Specifically:

1. A thinking-answering consistency (TAC) reward mechanism that enforces consistency between the chain of reasoning (thinking), the final answer (answering), and the ground truth (GT).

2. During training, if the KL divergence exceeds a threshold, the sample is labeled as a "dirty sample“ and excluded from gradient updates.

3. The difficulty of a sample is assessed by computing its distance from the reward distribution.

**Strengths:**

1. The paper is well-organized.

2. The paper demonstrates that the TACO method is effective.

**Weaknesses:**

**Writing-wise:**

1. The paper does not provide sufficient justification early on for the assumption that accuracy equates to stability.

2. The paper devotes considerable space to demonstrating that TACO improves accuracy, but offers insufficient discussion on stability, the motivation highlighted in the introduction.

**Method-wise:**

1. The paper proposes three techniques based on three forms of consistency. However, as shown in Figure 3, the TAC mechanism actually increases training instability, and TACO does not appear to significantly improve the training stability of GRPO.

2. There is a lack of visual evidence  (eg. Figure 3) illustrating how the MKS and ADS components contribute to training stability.

**Experiment-wise:**

1. The experiments lack ablation studies on hyperparameters.

2. Table 7 is missing results for the MKS variant.

3. Given that DAPO removes the KL divergence term, it would be valuable to include DAPO results in Table 1 to better demonstrate TACO’s effectiveness and the necessity of the KL divergence constraint.

4. Experiments with larger models (e.g., 7B parameters) are missing; adding them would help assess how TACO scales with increasing model capacity.

**Questions:**

See Weakness

**Details Of Ethics Concerns:**

There is concern that geographical disparities might lead to discriminatory data filtering, as the method takes into account differences between data distributions.

---

### Official Review · Reviewer_a545 · 2025-11-01

**Soundness:** 3
**Presentation:** 3
**Contribution:** 2
**Rating:** 4
**Confidence:** 3

**Summary:**

This paper identifies three main failure modes: exploration collapse, inefficient learning, and ineffective reasoning, characterized by logical inconsistencies between reasoning processes and outputs. To address these issues, the authors introduce TACO, a reinforcement learning framework designed to enforce multi-level consistency.

TACO consists of three integrated components:

- Think-Answer Consistency (TAC) Reward: Ensures alignment between reasoning, answers, and ground truth for semantic integrity.
- Memory-Guided KL Stabilization (MKS): Dynamically defers high-risk updates to prevent optimization collapse.
- Adaptive Difficulty Sampling (ADS): Optimizes data curation for more efficient learning.

Extensive experiments demonstrate TACO's superiority, achieving top performance across 15 benchmarks, including Referring Expression Comprehension (REC), Visual Question Answering (VQA), and long-horizon Video VQA. TACO shows enhanced generalization, sustained efficiency, and improved stability in long-chain reasoning, outperforming traditional reinforcement learning approaches.

**Strengths:**

1. The paper is well-organized, and the writing is clear.

2. The experiments include a variety of datasets, such as Video VQA, VQA, and REC, which cover a wide range of multimodal scenarios.

3. The main point of this paper is intriguing; it introduces the concept of consistency supervision between thinking and answering, which I find appealing. It would be even better to provide dense rewards at the semantic level or within a continuous space, rather than relying solely on rule-based answer rewards.

**Weaknesses:**

1. The paper claims that three failure modes arise from a systemic breakdown in consistency across semantic, optimization, and learning levels; however, if I do not miss something, the paper lacks the motivation and empirical evidence to support this perspective.

2. The paper focuses solely on comparing the RL method, such as GRPO, and proposes improvements to this approach. However, many other RL methods exist, including classic techniques like DPO and PPO, as well as GRPO variants such as DAPO and Reinforce++. If I’m not mistaken, the paper does not compare these methods or adapt TACO to them. It would be beneficial to include experiments on these alternatives.

3. While the method is based on VLM, it appears that it could be easily adapted to LLM tasks. What aspects of the method must be tied to VLM? Additionally, what is its performance on LLM tasks, such as GSM8K? I am uncertain whether these three failures also occur in LLM tasks.

If the authors can address some of my concerns, I will consider raising my score.

**Questions:**

See the weakness.

**Details Of Ethics Concerns:**

no concerns

---

### Official Review · Reviewer_p6LR · 2025-11-01

**Soundness:** 3
**Presentation:** 2
**Contribution:** 3
**Rating:** 4
**Confidence:** 4

**Summary:**

The paper proposes TACO, a RL-based training framework that aims to improve reasoning in Large Vision-Language Models by enforcing consistency at three levels: semantic (via Think-Answer Consistency, TAC), optimization (via Memory-Guided KL Stabilization, MKS), and learning (via Adaptive Difficulty Sampling, ADS). The framework is evaluated on multiple Referring Expression Comprehension (REC) and Visual Question Answering (VQA) benchmarks, where it demonstrates performance gains over the GRPO baseline.

**Strengths:**

1. The idea of enhancing RL training stability and robustness through MKS and ADS is practically useful.
2. This paper supports its claims regarding MKS and ADS with empirical evidence, including experimental logs during training in Fig. 3 and performance on various REC (Tab. 1) and VQA (Tab. 3 and 5) datasets.

**Weaknesses:**

[Major Weakness]
1. Although the paper claims to target ineffective reasoning, the TAC design feels less novel for VQA (since it only directly uses a LVLM as an reward model), and the TAC design for REC, especially how it captures reasoning consistency, remains vague (which would be further elaborated in 3.), weakening the conceptual appeal of the method.
2. The proposed MKS and ADS components are general RL stabilization strategies, not strictly tied to GRPO, and it remains unclear whether their benefits are specific to GRPO or would generalize to other RL algorithms such as PPO.
3. The formulation in Eq. 3 lacks clarity, since the roles of $BBox_1$, $BBox_2$, and $BBox_3$ are not explicitly defined. Additionally, the predicted bounding boxes from the qualitative examples in the appendix seem to be not aligned between thinking process and answers, raising doubts about the effectiveness of TAC.
4. It remains unclear whether the performance gain largely depends on TAC or whether combining MKS and ADS with GRPO alone also produces a similar boost; such an ablation is missing and would make the contribution of TAC more convincing.

[Minor Weakness]
1. The training and testing dataset used for long-horizon video VQA tasks (in Tab. 6) is not fully described, making it hard to assess reproducibility and significance.
2. In Tab. 7, the acronym “RRS” is likely a typo of “MKS” or an undefined term.

**Questions:**

1. When training on REC tasks, how does the model calculate the TAC reward if the thinking process does not include a bounding box? In that case, how is the reward signal handled?

---

### Note · Authors · 2025-11-14

I have read and agree with the venue's withdrawal policy on behalf of myself and my co-authors.